# Serotonin Exposure Improves Stress Resistance, Aggregation, and Biofilm Formation in the Probiotic *Enterococcus faecium* NCIMB10415

**Rossella Scardaci [1], Marcello Manfredi [2], Elettra Barberis [2], Sara Scutera [3], Emilio Marengo [2] and Enrica Pessione [1,*]**

[1] Department of Life Sciences and Systems Biology, University of Torino, 10123 Torino, Italy; rossella.scardaci@unito.it
[2] Department of Translational Medicine, Università del Piemonte Orientale, 28100 Novara, Italy; marcello.manfredi@uniupo.it (M.M.); elettra.barberis@uniupo.it (E.B.); emilio.marengo@uniupo.it (E.M.)
[3] Department of Public Health and Pediatric Sciences, University of Torino, 10126 Torino, Italy; sara.scutera@unito.it
[*] Correspondence: enrica.pessione@unito.it

**Abstract:** The role of the microbiota–gut–brain axis in maintaining a healthy status is well recognized. In this bidirectional flux, the influence of host hormones on gut bacteria is crucial. However, data on commensal/probiotics are scarce since most reports analyzed the effects of human bioactive compounds on opportunistic strains, highlighting the risk of increased pathogenicity under stimulation. The present investigation examined the modifications induced by 5HT, a tryptophan-derived molecule abundant in the intestine, on the probiotic *Enterococcus faecium* NCIMB10415. Specific phenotypic modifications concerning the probiotic potential and possible effects of treated bacteria on dendritic cells were explored together with the comparative soluble proteome evaluation. Increased resistance to bile salts and ampicillin in 5HT-stimulated conditions relate with overexpression of specific proteins (among which Zn-beta-lactamases, a Zn-transport protein and a protein involved in fatty acid incorporation into the membrane). Better auto-aggregating properties and biofilm-forming aptitude are consistent with enhanced QS peptide transport. Concerning interaction with the host, *E. faecium* NCIMB10415 enhanced dendritic cell maturation, but no significant differences were observed between 5HT-treated and untreated bacteria; meanwhile, after 5HT exposure, some moonlight proteins possibly involved in tissue adhesion were found in higher abundance. Finally, the finding in stimulated conditions of a higher abundance of VicR, a protein involved in two-component signal transduction system (VicK/R), suggests the existence of a possible surface receptor (VicK) for 5HT sensing in the strain studied. These overall data indicate that *E. faecium* NCIMB10415 modifies its physiology in response to 5HT by improving bacterial interactions and resistance to stressors.

**Keywords:** microbiota–gut–brain axis; stationary phase growth; ampicillin resistance; bile salt tolerance; gel-free proteomics; moonlight proteins; two-component systems



## 1. Introduction

The cross-talk between enteric bacteria and the human host often occurs by means of shared molecules that support reciprocal physiological regulation [1]. It is now clear that a common language exists between phylogenetically very distant organisms and some informational molecules that were conserved during evolution, are present in "brainless" organisms such as bacteria, where they generally perform a different function [2]. In the context of this bidirectional regulation, a bottom-up effect has been described, where host physiological functions such as immunity, appetite, sleep, and mood can be controlled by bacterial-derived molecules [3,4]. Some examples regard bacteria as able to produce brain-targeting neuroactive compounds such as gamma-amino butyric acid [5,6]

dopamine and nor-epinephrine [7], melatonin [8], indole, similar to melatonin [9], a muramyl dipeptide similar to serotonin [10], and serotonin (5HT) [11]. In parallel, bacterial traits such as pathogenicity, toxin production, ability to form biofilm and to aggregate, as well as to tolerate stressing conditions can be modulated by host-derived compounds by a top-down control system [12,13]. In particular, prokaryotes, especially pathogenic bacteria, can respond to host-derived compounds such as dopamine [14], norepinephrine (NE) [15,16], epinephrine [17], melatonin [18], and 5HT [19], often by enhancing their pathogenic potential.

The importance of tryptophan and its metabolites in the microbiota gut–brain–axis has been highlighted in the works of O'Mahony [20] and Galligan [21]. This amino acid can be directly synthesized by enteric bacteria possessing tryptophan synthase [22,23], or can derive from food. Therefore, its availability depends on the microbiota composition and the diet. In addition, it can be channeled in different metabolic pathways by gut bacteria, among which the most important are those generating indole, 5HT, and melatonin [24]. In particular, 5HT can be produced from tryptophan by different bacterial genera, including *Morganella*, *Klebsiella* [11], and lactic acid bacteria (LAB) (*Streptococcus*, *Lactobacillus, and Lactococcus)* [25,26], and also by enteroendocrine (EEC) and enterochromaffin cells (ECC) in the human gut mucosa by using both diet-derived and microbial synthesized tryptophan.

On the other hand, it must be underlined that 90% of human 5HT is produced in the gut, where it is involved in GI motility, secretion, and initiation of vagal and nociceptive reflexes through the activation of diverse families of 5-HT receptors. Moreover, alterations in 5HT signaling in the gut have been linked to the pathophysiology of several GI disorders [27].

In this context, it is of high interest to evaluate how bacteria can respond to changes in the gut environment due to sudden increase of 5HT. In the literature, data on the biological effects of this molecule on prokaryotes are scarce in comparison with other hormones (i.e., epinephrine and NE), and most reports concern only pathogenic bacteria [19,28].

In the present work, we take into consideration the effects exerted by 5HT on a particular commensal/probiotic strain of LAB, *E. faecium* NCIMB10415. Most Enterococci belonging to human microbiota display beneficial features, and therefore could be classified among probiotics. However, for the Enterococcus genus, some controversial behavior is reported concerning virulence factors, production, and antibiotic resistance [29]. These unwanted characters could be modulated by external environmental factors; therefore, it is of high importance to assess safety in every pathophysiological condition that may occur in the human gut. Analyses of some phenotypic traits, such as growth rates, auto-aggregation, biofilm formation, resistance to bile salts, and antibiotic response profiles, evaluated in stimulated and control conditions, revealed that 5HT can alter *E. faecium* NCIMB10415 physiology. Furthermore, the gel-free proteomic results are consistent with the observed modifications, also indicating the presence of a receptor-like protein for serotonin in this bacterial strain.

## 2. Materials and Methods

### 2.1. Bacterial Strain and Growth Conditions

*E. faecium* NCIMB 10415, isolated from a probiotic preparation, was stocked at −20 °C in 25% glycerol. A rich chemically defined medium (CDM) [30,31] at pH 7 was selected for culturing bacteria in control condition and treated with serotonin (serotonin hydrochloride, Sigma-Aldrich Inc., St. Louis, MO, USA) in order to avoid the differences between commercial batches and to easily replicate the exact composition in the laboratory, weighing every nutrient. The components of CDM are presented in Supplementary Materials and Methods (SM1). Brain heart infusion (BHI, (Sigma-Aldrich Inc., St. Louis, MO, USA) supplemented with 1.5% agar was used as solid medium (BHI/A,Sigma-Aldrich Inc., St. Louis, MO, USA). 5HT was freshly dissolved in CDM, filter sterilized, and diluted accordingly to the experiments.

## 2.2. Growth Kinetics

Growth kinetics were recorded in a multiplate reader (Filtermax F5) comparing the control curves in CDM to the treatments with multiple concentrations of 5HT in CDM (0.5 nM, 500 nM, 0.5 μM, 50 μM, 500 μM). Orbital shaking and reading $OD_{595}$ every 30 min for 24 h at 37 °C were the parameters chosen for the experiments. Each concentration was tested for 10 different wells, and the test was repeated three independent times. The error bars for each point of the curve correspond to the standard error for each set of measurements. The growth rates (μ) were calculated and expressed as % of the control ± standard deviation.

## 2.3. Phenotypic Evaluations

### 2.3.1. Resistance to Bile Salts

To test the resistance to bile salts, 1 mL of late exponential phase *E. faecium* culture (≈4 h of growth), treated with 50 μM 5HT or in control condition, was harvested by centrifugation (10,000× *g*, 10 min, RT) and the pellets were suspended in fresh CDM supplemented with 4.5% bile salts and no 5HT (bile, bovine, Sigma-Aldrich Inc., St. Louis, MO, USA). Then, the bacteria were incubated at 37 °C, for 4 h and 100 rpm, to mimic the intestine transit. The samples collected at T0 and T4 were diluted in 0.9% NaCl and counted on BHI/A after 24 h, 37°C. The bile salt tolerance (BST) was calculated as a survival rate (SR): $[CFU/mL\ T_4]/[CFU/mL\ T_0] \times 100$ [32], and the results are expressed as % of control.

### 2.3.2. Auto-Aggregation

The auto-aggregation in control bacteria and in presence of 5HT was executed as described previously [33] with some modifications. Concisely, cultures grown until the late exponential phase (4 h) were centrifuged (10,000× *g*, 10 min', RT) and suspended in NaCl 0.9%. Bacteria were then diluted to reach an $OD_{600}$ of about 0.3, and 1 mL was distributed in 10 sterile cuvettes. $OD_{600}$ was measured at $T_0$ ($OD_0$) and after 2 h at 37 °C ($OD_2$) straight into the cuvette. The auto-aggregation percentage was determined with the following formula: $[1-(OD_2/OD_0) \times 100]$. The results are a mean of three different experiments performed in three independent days.

### 2.3.3. Biofilm Formation

Whether the presence of 5HT might have influenced the formation of biofilm biomasses was investigated with the crystal violet method [34], with slight modifications. O/N precultures were diluted to an $OD_{600}$ 0.4 with or without 5HT in a 96-well plate and let to grow for 24 or 48 h at 37 °C, in the latter case with medium and molecule replacement after the first 24 h. At the end of the incubation period, planktonic cells were removed, and the plate was washed three times with MilliQ water. Cells were stained with 150 μL of 0.1% crystal violet (PanReac-AppliChem ITW Reagents, Glenview, IL, USA) for 15min at RT, the plate was then rinsed three times with MilliQ and let to dry perfectly at 37 °C; 150 μL of 99% EtOH was used to extract the crystal violet and the $A_{595}$ was measured (Filtermax F5). Results are expressed as percentages of control: $(A_{595}\ 5HT)/(A_{595}\ C) \times 100$.

### 2.3.4. Antibiotic Susceptibility Test

The possible modifications in the susceptibility to ampicillin (Amp) and vancomycin (Van) induced by 5HT on our strain were investigated by means of the E-test system (E-strip, BioMérieux Inc., St. Louis, MO, USA). Briefly, bacteria cultured in control condition and with 5HT were harvested at the early stationary phase, 100 μL of cultures were then spread on BHI/A and the antimicrobial agent strips were applied to the plates. After incubation (O/N, 37 °C), a zone of growth inhibition was seen around the plates, and the minimal inhibitory concentration (MIC) was read from the scale on the strip.

### 2.4. Whole Cell Proteomic Analyses

2.4.1. Soluble Proteins Extraction

A volume of culture corresponding to 50 mg of dry weight of cells was collected in late exponential phase (4 h of growth) by centrifugation at $4000 \times g$ for 20 min at 4 °C, for each sample of control and 50 μM 5HT-treated bacteria. Pellets were washed twice with NaCl 0.9% and resuspended in 3 mL of Tris-HCl 50 mM, 1 mM EDTA, pH 7.3, and sonicated on ice for 30 min at 20 KHz with intervals of 20 s. To obtain the highest amount of proteins, the unbroken cells were recovered by centrifugation at $4000 \times g$ for 20 min at 4 °C, and the procedure was repeated on the pellets [35]. Six milliliters of supernatant was then ultracentrifuged at $18,000 \times g$ for 45 min at 4 °C in a Beckman L8-60M Ultracentrifuge, Ti60 rotor [36]. Proteins were precipitated with chloroform and methanol according to Wessels and Flügge [37]. The pellet containing only proteins was resuspended in 500 μL of solution containing 50 mM $NH_4HCO_3$. The protein quantification was assessed by 2-D Quant kit (GE Healthcare, Little Chalfont, Buckinghamshire, UK). The protein samples were collected from three biological replicates.

2.4.2. In-Solution Protein Digestion

Prior to SWATH-MS (sequential window acquisition of all theoretical fragment ion spectra mass spectrometry) [38,39], proteins were digested in trypsin. An amount of 100 μg of protein in 25 μL of 100 mM $NH_4HCO_3$ was reduced with 2.5 μL of 200 mM DTT (Sigma-Aldrich Inc., St. Louis, MO, USA) at 90 °C for 20 min and next alkylated with 10 μL 200 mM iodoacetamide (Sigma-Aldrich Inc., St. Louis, MO, USA) for 1 h at RT and protected from light. Any excess of iodoacetamide was removed by the addition of 200 mM DTT [40]. The samples were then diluted with 300 μL of MilliQ and 100 μL of 100 mM $NH_4HCO_3$ to raise the pH to 7.5/8, and 5 μg of trypsin (Promega, Madison, WI, USA, Sequence Grade) was added. After an ON incubation at 37 °C, 2 μL of neat formic acid stopped the trypsin activity and digests were dried by speed vacuum [41]. The peptide digests were desalted on the Discovery® DSC-18 solid phase extraction (SPE) 96-well plate (25 mg/well) (Sigma-Aldrich Inc., St. Louis, MO, USA) as reported elsewhere [42].

2.4.3. SWATH-MS Analysis

LC–MS/MS was executed on a micro-LC Eksigent Technologies (Dublin, CA, USA) system, using the Halo Fused C18 column (0.5 × 100 mm, 2.7 μm; Eksigent Technologies, Dublin, CA, USA) as stationary phase. A volume of 4.0 μL was injected every time and the oven temperature was 40 °C. 0.1% (*v/v*) formic acid in water (A) and 0.1% (*v/v*) formic acid in acetonitrile (B) were mixed at increasing concentrations of B from 2% to 40% eluting at a flow rate of 15.0 μL/min for 30 min. A 5600+ TripleTOF system (AB Sciex, Concord, ON, Canada) equipped with a DuoSpray Ion Source and CDS (calibrant delivery system) was put at the interface with the LC system. The samples utilized to produce the SWATH-MS spectral library were the first object of the traditional data-dependent acquisition (DDA) and afterwards, for the label-free quantification of the cyclic data independent analysis (DIA) of the mass spectra, as previously described [43]. The MS data were acquired with Analyst TF 1.7 (SCIEX, Concord, ON, Canada). Three instrumental replicates for each of the three biological replicates were subjected to the DIA analysis [44].

2.4.4. Protein Data Search

The software Protein Pilot v. 4.2 (SCIEX, Concord, Vaughan, ON, Canada) was used to search the MS files. Cysteine alkylation, digestion by trypsin, no special factors, and false discovery rate at 1% were set up as parameters. The files were additionally examined with Mascot v. 2.4 (Matrix Science Inc., Boston, MA, USA) using trypsin as enzyme, 2 missed cleavages, and a search tolerance of 50 ppm set for the peptide mass tolerance, 0.1 Da for the MS/MS tolerance, charges of peptides to search for were specified with 2 +, 3 +, and 4 +, and the search was set on monoisotopic mass. The instrument was set to ESI-QUAD-TOF and the modifications for the search were set as follows: carbamidomethyl

cysteines as fixed modification and oxidized methionine as the variable one. The DIA files were subsequently switched to pseudo-MS/MS spectra by the DIA-Umpire software and were searched as DDA files on Mascot and Protein Pilot using the same parameters used for the DDA search. The UniProt/Swiss-Prot reviewed database containing *E. faecium* proteins (NCBI_Enterococcus_Faecium, version 01042019, 12110 sequence entries) was utilized.

### 2.4.5. Protein Quantification

PeakView 2.0 and MarkerView 1.2. (Sciex, Concord, ON, Canada) were used to obtain a label-free quantification of the ion chromatograms of all unique ions for the resulting peptides. The DDA acquisitions and a protein FDR threshold of 1% were utilized to build an integrated assay library. Six transitions per peptide and six peptides per protein were extracted from the SWATH files, excluding peptides with modifications and shared peptides. *T*-test was performed on peptides with FDR < 1% exported in MarkerView. A *p*-value < 0.05 and fold change > 1.3 were selected as maximum values to choose proteins with lower or higher abundance. The data concerning the mass spectrometry proteomics, with the dataset identifier PXD023456, are deposited in the ProteomeXchange Consortium by the PRIDE [45] partner repository (https://www.ebi.ac.uk/pride/archive/, accessed on 7 January 2021).

### 2.4.6. Protein Classification

All the annotated proteins differentially abundant in the two conditions were analyzed using the UniProt/Swiss-Prot reviewed database (https://www.uniprot.org/, accessed on 7 January 2021) and the eggNOG database of orthologous groups and functional annotation (http://eggnogdb.embl.de/#/app/home, accessed on 7 January 2021). Clusters of orthologous groups (COGs) of proteins are provided by the eggNOG resource [46] that represents the base of a functional protein classification based on precisely deciphered evolutionary relationships [47].

### *2.5. Immune-Stimulating Activity of Killed E. faecium NCIMB 104145 Cells and Cell Free Supernatants*

### 2.5.1. Preparation of the Bacterial Strain and Cell-Free Culture Supernatant for Dendritic Cells (DC) Maturation Assays

E. faecium NO SPACENCIMB104145 was cultured in the presence or absence of 50 μM 5HT at 37 °C for 4 h (late exponential phase). Bacterial cells were harvested by centrifugation (12,000× *g*, 10 min, RT), washed twice in sterile PBS and resuspended in the same buffer. For the preparation of the heat-killed (HK) samples, bacterial cells were heated for 30 min at 90 °C. Complete loss of cell viability was verified by monitoring colony formation on agar plate.

### 2.5.2. Monocyte-Derived DC Preparation

Monocytes were isolated from peripheral blood mononuclear cells obtained from healthy donor buffy coats (through the courtesy of the S.C. Centro Produzione e Validazione Emocomponenti, Torino) by immunomagnetic selection with CD14 microbeads (MACS monocyte isolation kit from Miltenyi Biotec, Bergisch Gladbach, Germany). This procedure yields a greater than 98% pure monocyte population, as assessed by fluorescence-activated cell sorter analysis (FACSCalibur, BD Biosciences, Franklin Lakes, NJ, USA). To obtain monocyte-derived DCs, monocytes were cultured for 5 days at $10^6$ cells/mL in RPMI 1640 medium (Gibco Thermo Fisher Scientific, Waltham, MA, USA) containing 10% FCS in the presence of GM-CSF (50 ng/mL) and IL-4 (20 ng/mL) (both from PeproTech, Rocky Hill, NJ, USA).

### 2.5.3. Monocyte-Derived DC Stimulation

Immature DC were seeded at the density of $1 \times 10^6$/mL and incubated with HK Enterococcus faecium NCIMB 10415 (bacteria/host ratio 10:1), with CSF added to the DC

culture medium at a concentration of 10–30% $v/v$ and LPS (100 ng/mL) for 40 h. Cells and cell supernatants were collected for flow cytometric and cytokine analysis.

### 2.5.4. Flow Cytometric Analysis

DC were acquired with a FACSCalibur and analyzed using Flowlogic software (Miltenyi Biotec, Bergisch Gladbach, Germany), using the following antibodies: anti CD80-PE, anti CD-83-VioFITC, and the corresponding isotype control antibodies (all purchased from Miltenyi Biotec, Bergisch Gladbach, Germany).

### *2.6. Statistical Analyses*

Data collected from a minimum of three experiments are expressed as mean ± standard error (SEM) and statistical significance assessed using Student's *t*-test using the software GraphPad Prism 6.

## 3. Results

The possible physiological modifications occurring in *E. faecium* NCIMB10415 after treatment with 5HT were assessed with references to the soluble proteomic profiles. Then, the effect of HK treated and untreated enterococci on DC was investigated to elucidate possible feedback modulations on human immune cells due to 5HT stimulation of the bacterial strain.

### *3.1. Growth Kinetics*

Figure 1 illustrates the growth curves of *E. faecium* NCIMB10415 stimulated with different doses of 5HT (0.5 nM, 50 nM, 500 nM, 50 μM, and 500 μM) including control conditions (no 5HT). Higher $OD_{595}$ values after treatments with all the different concentrations of the molecule are perceivable since the beginning of the stationary phase ($\approx$4 h of growth), but the difference becomes more evident during the middle and late stationary phase (namely after 15 h of growth). Precisely, the $OD_{595}$ at 24 h for the 0.5 nM, 50 nM, 500 nM, 50 μM, and 500 μM treatments are 1.0 ± 0.13, 1.1 ± 0.11, 1.3 ± 0.12, 1.2 ± 0.01, and 1.1 ± 0.02-fold greater than untreated controls, respectively (SM3). SM3 also presents the CFU counting at 24 h for each concentration of 5HT. The growth rates ($\mu$), expressed in percentage of control, are only slightly increased by the hormone and are in order: 103.2 ± 1 %, 104.3 ± 1.3 %, 104.9 ± 0.5%, 103.6 ± 1.5 %, and 101.3 ± 0.8 %. Among all, the concentration of 50 μM 5HT was selected to treat *E. faecium* NCIMB10415 for the successive studies, as justified in the following paragraphs.

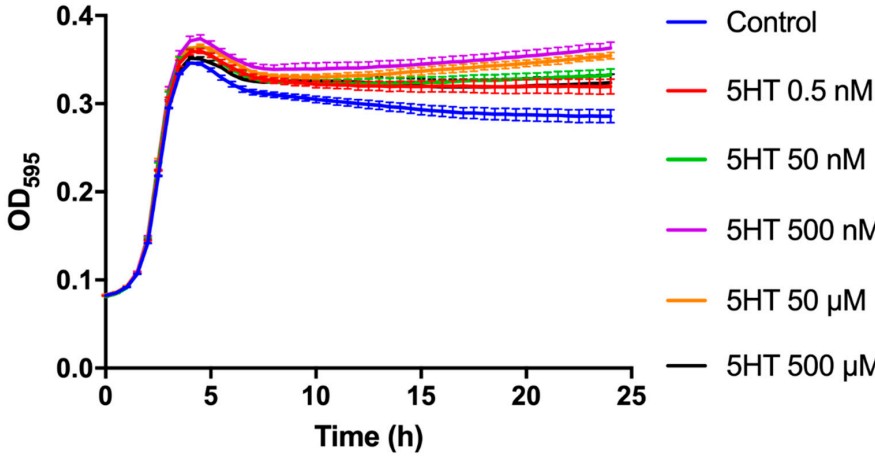

**Figure 1.** Growth changes. Growth kinetics of 24 h: effect of different concentrations of 5HT (0.5 nM, 50 nM, 500 nM, 50 μM, 500 μM) on the growth of *E. faecium* NCIMB10415. The effect of the molecule is detectable during the mid and late stationary phase.

### 3.2. Phenotypic Evaluations

3.2.1. Bile Salt Resistance

We investigated if 5HT treatment could modify the ability of *E. faecium* NCIMB10415 to tolerate the surfactant and membrane-damaging activity of bile salts by comparing the final cell number of 5HT-exposed and unexposed bacteria and calculating the BST. In Figure 2, the results obtained clearly indicate that 50 μM 5HT treatment increased our strain's rate of survival of approximately 40%.

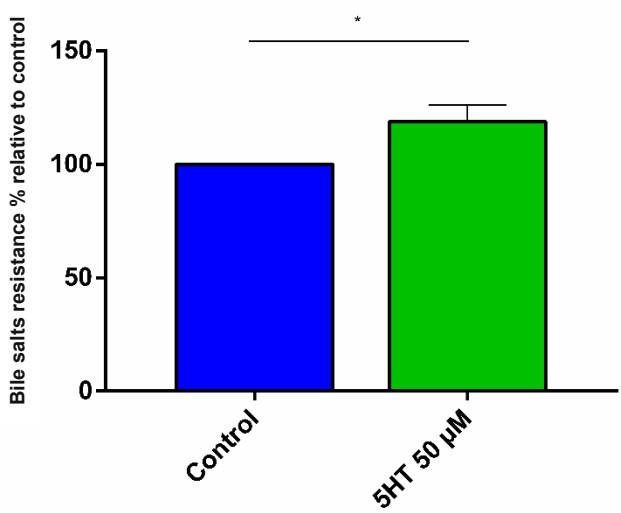

**Figure 2.** Bile salt resistance. Bile salts (4.5%) tolerance of *E. faecium* NCIMB10415 is expressed as % of control and is increased after a pre-treatment with 50 μM 5HT (* $p < 0.05$).

3.2.2. Auto-Aggregation Assay

Bacterial auto-aggregation, as an index of good gut colonization ability, was evaluated. The results reported in Figure 3 reveal an increased auto-aggregation capability of *E. faecium* NCIMB10415 (+25%) when treated with 5HT; precisely, the percentages of auto-aggregation were $9.6 \pm 0.6$ for the control and $12.1 \pm 0.8$ for the treated samples.

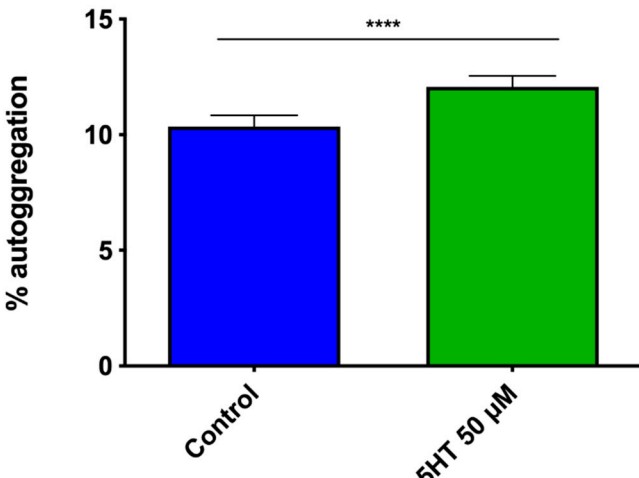

**Figure 3.** Auto-aggregation. Percent auto-aggregation of *E. faecium* NCIMB10415 is enhanced after the treatment with 50 μM 5HT (**** $p < 0.0001$) compared to the control condition.

3.2.3. Biofilm Formation

The aptitude of bacteria to persist in the gut environment, measured as biofilm-forming tendency, was tested in stimulated and control conditions. Here, we examined

how the exposure to 50 μM 5HT could affect the capability of *E. faecium* NCIMB10415 to form biofilm after 24 and 48 h of incubation. The difference in the biofilm biomasses of exposed and unexposed bacteria after the first 24 h of incubation is undetectable (Figure 4A), whereas it is clear and relevant at the end of the 48 h, after the removal of the old CDM and the addition of fresh medium and 5HT (Figure 4B). As represented in the bar chart, a 20% biofilm biomass enhancement after 48 h is detectable in the cultures stimulated with the human hormone compared to the control condition.

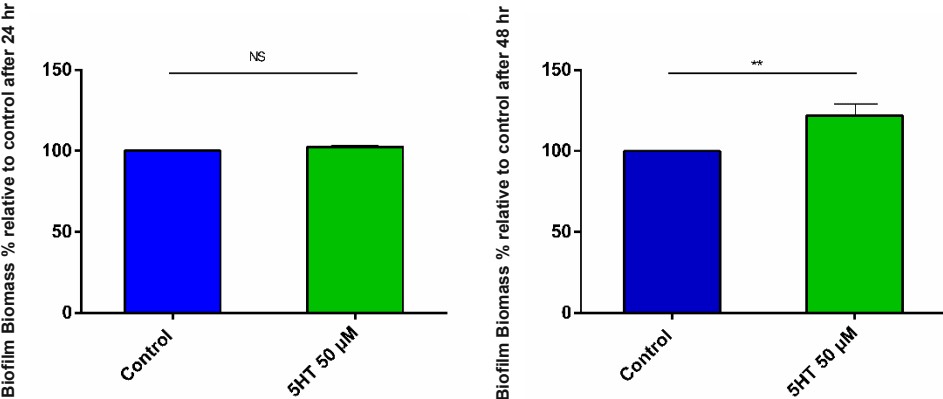

**Figure 4.** Biofilm. Biofilm biomass formation revealed with the crystal violet staining method, after the treatment of *E. faecium* NCIMB10415 with 50 μM 5HT for 24 h or 48 h, and expressed in % of control. The stimulating effect of 5HT is detectable only after 48 h of growth (NS: not significant, ** $p < 0.01$).

### 3.2.4. Antibiotic Susceptibility

The antibiotic susceptibility test was intended to measure whether the pretreatment with 5HT could affect the susceptibility of *E. faecium* NCIMB10415 to Amp and Van. The MIC increased almost 50% for Amp ($0.38 \pm 0.00$ μg/mL for control, $0.56 \pm 0.10$ μg/mL for 5HT treated) as reported in Table 1, while the variation in MIC for Van was not significant.

**Table 1.** Antibiotic susceptibility. *E. faecium* NCIMB10415 treated with 50 μM 5HT modifies its sensitivity to Amp relative to control condition (** $p < 0.01$) but not to Van (ns: not significant).

| Treatment | MIC Amp (μg/mL) | MIC Van (μg/mL) |
|---|---|---|
| Control | $0.38 \pm 0.0$ | $1.1 \pm 0.3$ |
| 5HT 50 μM | $0.56 \pm 0.1$ [**] | $1.0 \pm 0.3$ [ns] |

### 3.3. Whole Cell Proteomic Analyses

SWATH-MS approach was utilized in order to identify possible 5HT-induced proteins or differently abundant housekeeping enzymes, combining high levels of protein recovery and reproducible data [38,39]. The analyses lead to the identification of 690 of the annotated proteins [48], listed in Supplementary Material 2 (SM2), and 45 were found as statistically differently abundant ($p$-value < 0.05), being at least 1.3-fold higher in one condition compared to the other. Only the 20% of these proteins showed a lower abundance after 5HT treatment, while 80% were more abundant. The COGs and the UniProt/Swiss-Prot reviewed database (www.uniprot.com, accessed on 7 January 2021) were utilized to assess and examine their biological functions so that the most notable proteins identified were divided in six functional categories, reported in Figure 5. Signal transduction (vicR, HMPREF0351_12362, fold change 1.4; MCP, HMPREF0351_12008, fold change 1.5; gdhA, HMPREF0351_11739 fold change 1.4). DNA replication (dnaN, HMPREF0351_10002, fold change 1.4; HMPREF0351_10671, fold change 1.5) and Nucleic acids metabolism (thiD, HMPREF0351_11067, fold change 1.6). Transport systems (HMPREF0351_11100, fold change

2.1; brnQ, HMPREF0351_10149, fold change 1.9; HMPREF0351_12100, fold change 2.2; HMPREF0351_12278 fold change 1.5; oppF, HMPREF0351_10097, fold change 1.3), moonlight proteins (arcB, HMPREF0351_11693, fold change 1.4; pdhA, HMPREF0351_11225, fold change 1.6; HMPREF0351_10049, fold change 2.4), stress/stress counteracting proteins (HMPREF0351_10014, fold change 2.2; uspA, HMPREF0351_10246, fold change 1.3; HMPREF0351_12544, fold change 2; HMPREF0351_10245, fold change 1.5), and proteins involved in resistance to antibiotics (HMPREF0351_10757, fold change 1.8; ppiA, HMPREF0351_10611, fold change 0.7).

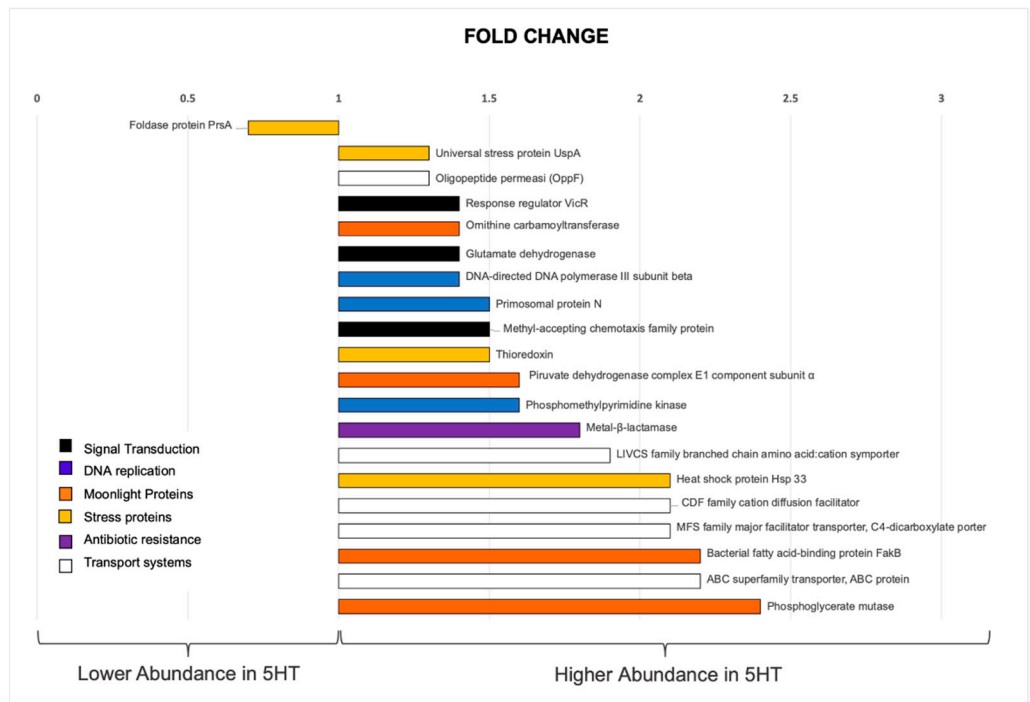

**Figure 5.** Proteomic relevant results. The bar chart shows the proteins with different abundance in 50 μM 5HT-treated *E. faecium* NCIMB 10415 compared to control condition (*p*-value $\leq 0.05$) and ordered by fold change. The proteins are divided in six main categories (signal transduction, DNA replication, moonlighting proteins, stress response proteins, proteins devoted to membrane transport, and to antibiotic resistance). On the left, the peptidyl-prolyl isomerase protein PrsA is displayed in lower abundance, while on the right, all the proteins with higher abundance in 5HT are presented. They are in order: Universal stress protein UspA, Oligopeptide permease (OppF), Response regulator VicR, Ornithine carbamoyltransferase, Glutamate dehydrogenase, DNA-directed DNA polymerase III subunit beta, Primosomal protein N, Methyl-accepting chemotaxis family protein, Thioredoxin, Piruvate dehydrogenase complex E1 component subunit α, Phosphomethylpyrimidine kinase, Metal-β-lactamase, LIVCS family branched chain amino acid: cation symporter, Heat shock protein Hsp 33, CDF family cation diffusion facilitator, MFS family major facilitator transporter, C4-dicarboxylate porter, FakB bacterial fatty acid-binding protein, and an ABC superfamily transporter.

### 3.4. 5HT Treated and Control E. faecium NCIMB10415 Effects on DC Maturation

We analyzed whether living cells of *E. faecium* NCIMB10415 untreated or treated with 50 μM 5HT differentially modulated the expression of maturation markers in DCs. We analyzed the CD80 and CD83 expression in terms of mean fluorescence intensity (MFI) using LPS as a positive control for DC maturation. The experiment performed with living cells was unsuccessful probably because the acidification produced during enterococcal lactic fermentation damaged the DC. Therefore, we decided to use heath killed (HK) enterococcal cells. The DC were incubated with HKE, HKE/5HT, and LPS, and evaluated for the expression of CD80 and CD83 as activation/maturation markers. We observed

an upregulation of CD80 following HKE and HKE/5HT stimulation, although at a lesser extent than with LPS. Concerning CD83, the maturation induced by HKE, HKE/5HT, and LPS were comparable. For both markers, the 5HT treatment did not induce significant differences relative to the untreated bacteria (Figure 6A,B).

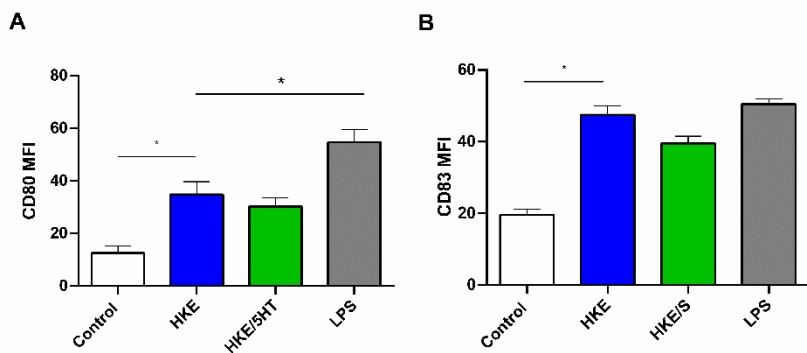

**Figure 6.** DC differentiation. CD80 (**A**) and CD83 (**B**) expression measured as mean fluorescence intensity (MFI), in DC untreated (Control), stimulated with LPS (positive control), or with 5HT-treated and untreated *E. faecium* NCIMB10415 (HKE/5HT and HKE) (* $p < 0.05$).

## 4. Discussion

Recently, a growing interest in the microbial responses to human-derived hormones has spread in the scientific community, mainly concerning pathogenic/opportunistic bacteria [49]. In a previous paper, we evaluated the phenotypic and proteomic modifications occurring in the probiotic *E. faecium* NCIMB 10415 after NE stimulation [13]. In the present research, the same strain was investigated to assess which physiological parameters especially important for the probiotic potential of the strain were affected by the treatment with 50 µM 5HT. We also analyzed the soluble protein profile modifications induced by the hormone, by using a cutting-edge comparative gel-free proteomic approach. The aim of the study was to ascertain if 5HT, a hormone mediating relatively different effects on human physiology [50] as compared to NE, could have similar or different effect on bacterial phenotypes.

The main findings support the following considerations: (i) *E. faecium* NCIMB 10415 can sense and respond to 5HT; (ii) a better stationary phase survival and improved resistance to various stressors, including Amp, was demonstrated; (iii) 5HT can induce a higher capability of auto-aggregation and biofilm formation and a higher expression of proteins mediating bacteria–bacteria and bacteria–host interactions. However, 5HT treatment did not affect the ability of *E. faecium* NCIMB 10415 to induce the expression of the activation markers CD80 and CD83 on dendritic cells.

Here, we try to discuss some crucial phenotypic aspects that vary after 5HT exposure relating them to the most interesting differentially abundant proteins, by adding some comments on their physiological roles. As far as other metabolic pathways are concerned, it is difficult to assess true stimulation or negative modulation. As an example, proteomic data show that some proteins of the route for cell-wall biosynthesis are more abundant (isomerase, epimerase) whereas others (transpeptidase) are in lower amount during 5HT exposure. Similar considerations can be done for ribosomal proteins.

### 4.1. 5HT Sensing

*Enterococcus faecium* NCIMB 10415 can sense and respond to environmental 5HT as an informational molecule, and in its presence, alterations of the growth pattern were observed, together with an improved capability to auto-aggregate, to form biofilm and to survive in the presence of bile acids and Amp. It is not clear at present whether the hormone can be uptaken by the cell or rather triggers phenotypic effects by means of a surface-bound receptor. However, the finding of a higher abundance of a peculiar protein,

the DNA-binding response regulator VicR, strongly supports the idea of the presence of a membrane sensor for 5HT in *E. faecium* NCIMB 10415. As a matter of fact, VicR belongs to the two-component transduction system (TCS) VicKR (YycFG), previously described in *E. faecalis* [51]. For the same species, other authors [52] suggested its role for sensing NE and modelized it by molecular docking. In general, signal transduction plays an important role in all living organisms, but it is fundamental in bacteria, which can sense rapidly changing environmental stimuli and respond accordingly. TCSs are the most spread systems allowing adaptation to environmental changes, such as variations of nutrients, communication signals, and toxic molecules [53], even if other mechanisms have been described [54]. In particular, these systems include an integral transmembrane kinase (HPK) that acts as a sensor for the stimulus and phosphorylates, and a response regulator (RR), which mediates the signaling events resulting in gene transcription and targeted expression of specific phenotypic characters in response [55]. Therefore, the greater abundance of VicR observed after 5HT stimulation might indicate the presence of the transmembrane VicK receptor, whose sensing domain is displayed on the *E. faecium* NCIMB 10415 cell surface. VicR signal transduction includes some genes critical for cell-wall biosynthesis and biofilm formation [56], in agreement with our experimental results (see next sections).

The VicKR system displays a 37% homology with the QS-detecting system QseCB from *E. coli*, which also recognize nor-epinephrine as stimulus [57]. It is reasonable to hypothesize that this receptor is able to recognize hormones containing an aromatic ring, such as those derived from the amino acids tyrosine, phenylalanine, and tryptophan.

Apart from TCS, a methyl-accepting chemotaxis family protein (MCP) also proved to be more abundant after growth in 5HT. In general, MCPs are transmembrane receptors for monitoring extracellular carbohydrate concentrations. It has been suggested that, in the closely related lactic acid bacterium *Streptococcus pneumoniae*, the mechanism might be used to control the activity of the linked histidine kinase [58], thus strengthening the communication with the external environment. For this purpose, glutamate dehydrogenase, also involved in signal transduction [59], has been found more abundant in 5HT-stimulated conditions, suggesting that the hormone can trigger an enhanced perception of external stimuli.

### 4.2. Modification of the Growth Profiles

The influence of 5HT on bacterial growth pattern has been recently studied on several commensal strains of human GI tract [60]. However, the results are contrasting and species-specific. Here, we also describe that supplementation of the culture medium with 5HT affected the bacterial growth kinetics of *E. faecium* NCIMB 10415 in a dose-dependent manner. All of the tested concentrations of the molecule proved to be effective in enhancing the bacterial growth (Figure 1); nevertheless, 500 nM and 50 μM resulted to be the most effective, decreasing the death phase effect. This is not surprising, since hormones act at an optimal concentration level that is generally in the mean concentration range [61].

In agreement with the higher growth yield observed in stimulated cells, a higher abundance of proteins involved in DNA replication, as the DNA-directed DNA polymerase III subunit beta and the primosomal protein N, was observed in 5HT-treated *E. faecium* NCIMB10415 cells. Proteomic data also may suggest that the hormone enhances transport of solutes across the membrane, especially acids, amino acids, and oligopeptides, so that optimized nutrient acquisition and increased metabolism can support a certain degree of cell duplication, even in the harsh conditions of the stationary phase. The MFS family major facilitator transporter, C4-dicarboxylate porter, well characterized in both *E. coli* and *B. subtilis* [62], is responsible of the uptake of four-carbon dicarboxylic acids such as malate, fumarate, succinate, and the amino acid aspartate. This system may ensure improved carbon utilization with energy gain, which can promote better stationary phase survival. In addition, the LIVCS-family branched chain amino acid (BCAA) cation symporter is also more abundant in stimulated conditions. This symporter provides the uptake of leucine, isoleucine, and valine inside the bacterial cell using $Na^+$ or $H^+$ as co-transported ions [63].

The BCAAs represent about 20% of the total protein amino acids in other LAB such as *L. lactis* [64]. For bacteria that cannot synthesize them, it is very important to obtain amino acids by proteolysis and uptake [65]. We may speculate that, at the end of the logarithmic phase, BCAAs begin to be released by proteolytic action on proteins derived from dead cell autolysis. Therefore, a higher abundance of this transporter provides a higher availability of these amino acids that can favor late stationary phase growth when most nutrients are exhausted. Similar considerations can be made for a specific amino acid ABC superfamily ATP-binding cassette transporter that resulted in being 1.5 times more abundant in 5HT treatment.

The improved survival of *E. faecium* NCIBM 10415 during the mid and late stationary phase in stimulated conditions can also be supported by the higher abundance of proteins generating ammonia, and hence involved in pH buffering, such as ornithine transcarbamylase (OTC) and glutamate dehydrogenase (GDH). Indeed, they provide an enhanced resistance to acidic stress, particularly frequent when the cultures reach a stationary growth profile. In LAB, this mechanism is particularly efficient and ensures survival when lactic fermentation is still active by balancing acidity. In detail, ornithine transcarbamylase converts citrulline into ornithine plus carbamoyl phosphate, and belongs to the arginine deiminase pathway (ADI pathway), a route used by LAB to enhance their energy gain (ATP) and to obtain medium alkalization since two moles of ammonia are produced at each cycle [66]. Moreover, this pathway supplies building blocks for pyrimidine biosynthesis useful during DNA duplication. To this purpose, a phosphomethylpyrimidine kinase was also found more abundant after 5HT stimulation. Hence, the high abundance of the enzyme OTC under 5HT stimulation supports the idea that the hormone can enhance bacterial fitness, stimulating energy metabolism, supporting building blocks for cell duplication, and especially balancing intracellular pH, which is very critical during the late exponential and stationary phase. In the present research under stimulated conditions, glutamate dehydrogenase (previously mentioned for its role in signal transduction), which catalyzes the degradation of glutamate to 2-oxoglutarate and ammonia, can also contribute to alkalization.

### 4.3. Enhanced Resistance to Stressors

Two paradigmatic stressing events were analyzed in addition to the commonly occurring stationary phase and oxidative stresses: survival in the presence of bile salts and antibiotics (Amp and Van).

### 4.3.1. Bile Stress Resistance

Bile is a surfactant mixture of inorganic ions, bile salts, cholesterol, and fatty acids [67], and it performs an antimicrobial activity towards gut bacteria as it can disrupt the membrane architecture by solubilizing phospholipids [68]. The genus *Enterococcus* is considered resistant to bile [69] and curiously, Saito and coworkers (2014) showed that *E. faecalis* is able to incorporate in the membranes the fatty acids from bile, enhancing its resistance to their surfactant activity and improving the membrane fluidity [70]. The proteomic results obtained in this study show after 5HT exposure a higher abundance of FakB, a protein involved in lipid incorporation into the cytoplasmic membrane. This may explain how the pretreatment with 50 μM 5HT enhanced the natural tolerance of *E. faecium* NCIMB10415 to bile by about 20% relative to the control (Figure 3).

### 4.3.2. Antibiotic Stress Resistance

After 5HT exposure, enhanced Amp resistance was detected by phenotypic tests (Table 1), while no significant modifications in the sensitivity pattern were detected for Van. Amp, together with other penicillins and cephalosporins, belongs to the class of the beta-lactam antibiotics. The most common mechanisms by which bacteria develop beta-lactam antibiotic insensibility are the production of degrading enzymes (beta-lactamases) and the modifications of the penicillin-binding proteins (PBPs) [71]. Proteomic data seem

to suggest that both mechanisms could be responsible of increased resistance to Amp observed after 5HT stimulation. As a matter of fact, a metal beta-lactamase (Zn-dependent hydrolase) was found in higher abundance in treated conditions. This type of hydrolase is characterized by the presence of one or two zinc atoms in the catalytic center that act as cofactors for catalysis [72], and this relates also to the higher abundance (2.1-fold change) of the CDF family cation diffusion facilitator. These integral membrane proteins mediate the uptake and extrusion of divalent cations belonging to the first and second transition series. An important number of them are involved in $Zn^{++}$ uptake and efflux [73]; therefore, it is possible to hypothesize a link between the overexpression of this protein and the enhanced abundance of the Zn beta lactamase in 5HT-stimulated conditions.

As far as the modifications of the target are concerned, the peptidyl prolyl isomerase (PrsA) proved to be in lower abundance after 5HT treatment. The PPiases are a class of foldases that support the cis/trans isomerization of the peptide bonds [74]. In particular, PrsA, a PPiase anchored to the outer side of the cytoplasmic membrane of several Gram-positive bacteria, acts as a foldase of many secreted proteins [75,76], including a certain class of penicillin binding proteins (PBPs) in *B. subtilis* [77]. A correct folding of PBPs is essential to ensure their efficient binding to the antibiotic. Therefore, the lower abundance of PrsA observed under 5HT stimulation could cause incorrect or incomplete folding of PBPs, which result is its inability to link Amp. Furthermore, a correlation between the previously noted glutamate dehydrogenase (1.4-fold more abundant after 5HT stimulation) and resistance to cell-wall targeted antibiotics has been previously observed also in *B. subtilis* [78].

### 4.3.3. Stationary Phase-Related Stress

The better survival of 5HT-stimulated cultures during the late stationary is in agreement with the concurrent enhanced abundance of the universal stress protein A, UspA, which is related to growth arrest that represents an adaptation to stationary phase conditions [79]. Actually, at this point of the growth, many stressors are present, besides nutrient depletion. In addition, since cell harvesting (for proteomic evaluation) occurred during the late exponential phase, the higher abundance of UspA in 5HT-stimulated conditions suggests that treated enterococci can anticipate the occurrence of stationary phase stress better than controls, thus improving their survival [80]. 5HT might therefore trigger the setup of an adaptive prediction on *E. faecium* NCIMB 10415, as previously noted in *E. coli* by other authors [81].

### 4.3.4. Oxidative Stress

Other two stress-counteracting proteins such as Hsp33 and thioredoxin were overexpressed after 5HT exposure. The heat shock protein Hsp33 belongs to the cellular pool of chaperones that ensure correct protein conformation and folding protection. It is also involved in misfolded proteins repair and degradation when an exogenous stressing event occurs [82]. In LAB, because of its microaerophilic nature and high sensitivity to oxygen [66], this protein is conserved among all genera to counteract oxidative stress [83]. It has been reported that Hsp33 undergoes activation during oxidative stress by the release of a Cys-bound Zn and formation of two disulfide bridges between $Cys^{232}$ and $Cys^{234}$ and $Cys^{265}$–$Cys^{268}$. In this conformation, it becomes competent for preventing protein aggregation behaving as a chaperone [84].

Thioredoxin, together with the thioredoxin reductase and the pyridine cofactor NADPH, is targeted to specifically control oxidative stress. It bears two cysteine residues, which undergo oxidation during its catalytic cycle [85].

The abundance of all these stress-counteracting proteins in stimulated conditions suggests that 5HT can induce several defense mechanisms in *E. faecium* NCIMB 10415 that are useful to increase physiological resistance to various stressors.

*4.4. Better Interbacterial Interaction: Autoaggregation and Biofilm Formation*

The bacterial auto-aggregation is triggered after sensing the presence of cells belonging to the same species in the external environment. This perception leads to the formation of a substantial clump that constitutes the basal structure at the bottom of a developing ecosystem [86]. In probiotics, this cell-assembling ensures better survival in the hostile gut environment and higher protection toward the activity of the host-immune system, resulting in a longer persistence in the intestine [87]. The auto-aggregation enhancement observed in 5HT-treated bacteria (Figure 3) might be partly related to the lower abundance of PrsA in this experimental condition. Actually, previous research demonstrated that a low expression of this isomerase caused enhanced cell envelope hydrophobicity that leads to improved auto-aggregating capability in *Streptococcus mutans,* a strain taxonomically close to *Enterococcus* [88]. For the strain in study, it is worth mentioning that this protein proved to be in lower abundance also after NE treatment, in a condition in which auto-aggregation was enhanced [13], similar to what was observed in the present work.

Moreover, auto-aggregation has been described as the initial phase in the formation of biofilm [87], a cluster of bacteria living in a self-produced matrix that provides nutrient sharing and resistance to external stressors [89]. In the present investigation, the biofilm lifestyle seems to be favored in presence of 5HT (Figure 4B). This enhanced propensity to social life can be due to the increased growth yield observed after 5HT. Furthermore, the finding of a higher abundance of the oligo peptide permease OppF could relate this with quorum sensing (QS). Actually, this protein belongs to the ABC transporter family (previously mentioned) and mediates the uptake of oligopeptides up to five amino acids, whatever the side chain. It is well established that biofilm formation is under QS control, and since in Gram-positive organisms QS mechanism is mediated by small peptides [90], it possible to speculate that the higher abundance of this oligopeptide transporter can favor intracellular transport of interbacterial signaling molecules. Although specific QS signals transporters are often involved, in agreement with this statement, it has been reported that oligopeptide signals are imported through ABC transporters at least in the Gram-positive *Bacillus subtilis* [91]. If this is also true for *E. faecalis*, as suggested by Leonard and coworkers [92], the increased abundance of these transporters might finally result in enhanced internalization of QS molecules that promote the setup of the biofilm lifestyle under 5HT effect. Furthermore, as referred in the paragraph concerning 5HT sensing, the VicKR system could also be involved in QS [57] and biofilm formation [56] that ultimately improve the bacterial resistance to the gut harsh conditions [93].

*4.5. Possible Interaction with the Host*

Evaluating the interaction of *E. faecium* NCIMB 10415 with the mammalian host was out of the scope of the present investigation. We limited our attention to compare the capability of 5HT-stimulated versus control bacteria to induce maturation of human dendritic cells (DC) *in vitro*.

Other LAB such as *Lactobacilli* have been shown to interact with DC and to induce strain-specific effects [94]. More specifically, activation of human and murine DCs by *E. faecalis* was demonstrated [95]. However, few data are available about the effects exerted on the immune system by *E. faecium* [96]. Here, we tested both living and heat-killed cells (HK) since the experiment with the living cells damaged the DCs cultures. Killed cells of *E. faecium* NCIMB 10415 displayed the ability to enhance DCs maturation (CD80 and CD83) relative to control conditions, suggesting that the DCs-activating components detected are not proteinaceous molecules that would be denatured after thermal stress. Therefore, it is possible that the observed boosting effect on DCs maturation is due to other surface components, such as lipoteichoic acids, teichoic acids, and EPS, which are not sensitive to thermal denaturation. In detail, CD80 and CD83 maturation was improved by enterococcal stimulation (both HKE and HKE/5HT) relative to control conditions (no external stimulating agent), although for CD80 this activity was not as efficient as that triggered by LPS stimulation (Figure 6A,B). This is reasonable since the *E. faecium* NCIMB 10415 is a

probiotic strain, whereas LPS is generally derived from pathogenic Gram-negative bacteria. When specifically considering their biological functions, both CD80 and CD83 relate to T lymphocytes activation. In the literature, experiments describing the effects of probiotic *E. faecium* strains on the host immune system only refer to increased neutrophil phagocytosis and humoral immunity [96]. Therefore, no comparison with known data is possible. Even if 5HT-treated enterococci seemed to be slightly less effective in inducing DC maturation than untreated bacteria, the difference observed is not statistically significant. Hence, further experiments are necessary to better assess the effect of 5HT on the microbiota–immune system axis.

These results on the potential immune-modulating effects of HKE and HKE/5HT suggest that the light immune-stimulating capabilities of this probiotic are not so deeply altered by 5HT exposure. However, 5HT also triggers other modifications on the bacterial phenotype that can affect the overall host–bacteria interactions. Among these modifications, those improving microbial adhesion are worth discussion. Actually, some proteins displaying moonlight function (moonlight proteins, MPs) and involved in host interaction have been found in higher abundance after 5HT treatment. MPs perform more than one role in different cell compartments [97]. Generally, in bacteria, housekeeping enzymes when secreted can act both as adhesins for host tissues and as immune stimulators. Among MPs, pyruvate dehydrogenase is a multimeric moonlighting enzyme whose subunits can interact with the host extracellular matrix components, such as fibronectin, as described in *Lactobacillus plantarum* [98]. The alpha subunit of the E1 component (PDHA), here found more abundant in 5HT-treated cultures, was demonstrated to bind plasminogen in a study concerning *Mycoplasma gallisepticum* [99]. This component has also been referred to as one of the main immunogenic proteins in *Streptococcus*, since it displayed the capability to induce antibody production in fish [100]. It is reasonable to assume that exposure of this protein on the bacterial surface could mediate these immunogenic effects. Here, we did not evaluate antibody production and we did not detect significant differences in DCs maturation elicited by treated vs. untreated cultures. However, it is possible that the alpha subunit of the E1 component (PDHA), here overexpressed, is not present in the HKE samples, because it undergoes denaturation following the thermal treatment, as previously reported.

Among the MPs found in higher abundance in stimulated cultures, phosphoglycerate mutase (PGM) is a central metabolism enzyme displaying the highest variation in treated and control cultures (2.4-fold change). It is a glycolytic/gluconeogenetic enzyme with moonlighting behavior in *Bifidobacterium lactis* BI07 and *Lactococcus lactis* [101,102], whose additional role is plasminogen binding. A second protein specifically binding the extracellular matrix protein fibronectin is the ornithine transcarbamylase (already mentioned as buffering agent), a moonlighting enzyme described as an adhesin in the nonpathogenic *Staphylococcus epidermidis* [103], but also as immune-stimulating antigen as reported in *Clostridium perfringens* by Alam et al. [104]. Further experiments intended to assess possible enhanced adhesion to mammalian tissues and immune cells after 5HT treatment are necessary to prove the significance of these findings.

## 5. Conclusions

Taken together, the overall results obtained in the present research on *E. faecium* NCIMB 10415 after 5HT stimulation reveal some similarities and some differences with previous data obtained with the same strain after NE treatment [13]. In detail, better survival during the stationary phase, enhanced resistance to bile salts, and improved aptitude to form biofilm and to auto-aggregate, all detected here, were also triggered by NE treatment.

Among the stress-counteracting responses specifically induced by 5HT, is worth noting an increased resistance to Amp, probably mediated by enhanced transcription of Zn-dependent beta lactamase, but also by altered folding of PBPs. Moreover, some proteins

attenuating stationary phase stress (UspA), oxidative stress (Hsp33, thioredoxin), and pH-related stress (OTC, GD) were more abundant after 5HT treatment.

In general, the proteomic data suggest a stronger effect of 5HT, since the protein fold changes were higher than those reported after NE stimulation. Furthermore, the induction of a possible hormone sensor VicK, involved in QS peptides perception as well, absent after NE stimulation, was suggested by the presence of its cognate RR VicR, in the present study.

To conclude, it is interesting to underline that while in *E. faecium* NCIMB 10415 NE and 5HT trigger similar (although not identical) effects, in the human host they often behave differently, inducing opposite reactions. Actually, at a systemic level, NE triggers stress responses, whereas 5HT induces relaxation, mood improvement, and stress adaptation [1]. However, it must be highlighted that in humans, at the gut level, negative effects can also be mediated by high levels of 5HT, similar to what occurs with NE [27]. Here, both NE and 5HT seem to go in the direction of improving environmental resistance of the strain, thus ensuring that its probiotic effect can be better achieved in the gut, whatever the stimulating molecule present in this context. In this respect, 5HT proves to better enhance the fitness of *E. faecium* NCIMB 10415 than NE. Since in the literature data concerning the effects of human hormones on probiotic bacteria are scarce, this report can constitute a valuable starting point to assess probiotic efficacy in the fast-changing conditions present in the human gut.

**Supplementary Materials:** The following are available online at https://www.mdpi.com/article/10.3390/microbiolres12030043/s1, SM1: CDM composition; SM2: proteomic results, SM3: OD 24 h table; CFU counting.

**Author Contributions:** R.S.: conceptualization, investigation, data curation, writing—original draft, methodology, validation, visualization, and formal analysis. M.M. and E.B.: methodology, software, and validation. S.S.: investigation and visualization. E.M.: data curation and supervision. E.P.: conceptualization, supervision, writing, reviewing, editing, project administration, and funding. All authors have read and agreed to the published version of the manuscript.

**Funding:** This work has been supported by Ricerca locale 2019–2020 of University of Turin and PhD funding.

**Data Availability Statement:** Mass spectrometry proteomics data are deposited in the ProteomeXchange Consortium by the PRIDE partner repository (https://www.ebi.ac.uk/pride/archive/, accessed on 7 January 2021) with the dataset identifier PXD023456.

**Acknowledgments:** The authors are grateful to Roberto Mazzoli for helping during the restyling of the manuscript and to Cinzia Bertea for supplying laboratory material necessary to complete the study.

**Conflicts of Interest:** The authors declare no conflict of interest.

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
