# Peer review of "Serotonin Exposure Improves Stress Resistance, Aggregation, and Biofilm Formation in the Probiotic Enterococcus faecium NCIMB10415"

_2036-7481, doi:10.3390/microbiolres12030043_

Round 1
Reviewer 1 Report
This study by Scardaci et al titled “ Serotonin exposure improves stress resistance, aggregation and
biofilm formation in the probiotic Enterococcus faecium NCIMB10415” studied the effect of 5HT on the probiotic Enterococcus faecium NCIMB10415. Scardaci et al notice that under the presence of 5HT Enterococcus faecium NCIMB10415 showed better survivability during stationary phase, increased resistance to bile salts, improved attitude to biofilm formation and auto-aggregation. This is a well done study with appropriate research design.
Author Response
We thank the reviewer for the all the positive comments about our study
Reviewer 2 Report
This manuscript by Scardaci et al. aims to describe the response of E. faecium NCIMB10415 to 5HT (serotonin) in an effort to better understand how bacterial metabolism in the gut affects the gut-brain axis. They present proteomic data as well as phenotypic data from several in vitro experiments that support the hypothesis that E. faecium can respond to 5HT in ways that would be relevant to behavior in the gut (changes in protein expression, biofilm formation, survival in bile). I agree that the relationship between gut bacteria (native microbiome or probiotic) and hormones or other human signaling molecules is an important one to understand, and I think the experiments that were chosen are the correct ones to test some of these hypotheses.
The overall data is compelling, but I find that the major dataset (the proteomics) is a bit over-interpreted and that parts of the paper are too speculative. Additionally, the way some of the data is presented figures is confusing.
My major comments are as follows:
Overall comment on figures: Because many of the figures look similar, it would be helpful if the axis labels were more descriptive. Nearly all of the figures have % of control as an axis label and Control or 5HT 50 uM as data labels. This make it very difficult to figure out what experiment is being summarized just by looking at the figure. For example, it would be helpful if the y-axis labels in Figure 4 said “% biofilm relative to control at 24 hr” instead of simply “% of control”
Line 86: I am concerned about bacterial stocks being stored at -20 oC, as it is much better to have bacterial stocks stored at -70 or -80 oC.
Line 88: What was 5HT dissolved in? Was this solvent used as a buffer control during growth experiments?
Lines 104-111 (bile salts): were cells grown with 5HT the whole time (pre and post bile salts)? As written, it is unclear when the 5HT treatment occurred.
Line 125: Why were biofilm assays started at an OD of 0.4? It is standard protocol to use a 1:100 dilution of overnight cultures or an inoculum of ~10^7 CFU/mL for biofilm assays. Starting at 0.4 does not allow for many doublings, considering the max OD for E. faecium in many types of growth media is ~2.0. Starting at a high OD can also disrupt gene expression networks that occur at low density before quorum sensing mechanisms happen as cells approach higher ODs.
Figure 1: Are the differences in OD in late stationary phase statistically significant? Do the changes in OD actually represent a difference in viability? Many things (such as cell shape, autolysis, etc.) can affect the OD600 in a growth curve.
Figure 1: The colors are hard to differentiate because the shades are similar. Please consider using a more colorblind-friendly palette or using different data point shapes for these curves.
Figure 3: What does each data point represent? The methods section says the experiment was repeated 3 times over 3 days (so 3 biological replicates). Could the authors clarify how many data points are shown? The methods section suggests there might be 10 technical replicates for each biological replicate, but there appear to only be 19 or 20 data points.
3.2.3 (Biofilm formation): There was no effect on biofilm formation after 24 hours – could this be because cells were seeded at such a high OD (0.4)? Also, adding 5HT to growth medium increased growth as shown in Figure 1. Therefore, is it possible that there is an increase in biofilm activity after 48 hr because the cells are growing more? It is preferable to quantify both cell growth and stained biofilm material in these types of assays in order to determine this.
Lines 287-291 (Antibiotic susceptibility): was 5HT added to the plates? Or only to the liquid media as cells were growing? This section sounds like 5HT was added to the plates whereas the methods section makes it sound like the cells were pre-grown in 5HT, but that cells were not exposed to 5HT and the E-test strips simultaneously.
Figure 5: It is confusing to see MIC data presented as bar graph, and it would be much easier to interpret if actual values were presented in a table (the numbers presented in lines 287-290). This will also make it easier to compare these results to previous studies that have measured MIC values for this strain in the absence of 5HT.
Figure 5, legend: Panel A should be discussed before panel B.
Figure 6: Please consider using a colorblind friendly palette, as the red and green will be impossible to differentiate for some readers. Also, please consider adding the gene numbers or locus tags as labels for the bars as that will make it easier for readers to interpret.
Figure 7: The title is incorrect and should not say “Antibiotic susceptibility”
Overall comment about discussion section: Much of the discussion section seems to contain results (such as the fold change of several proteins of interest). It would be better to discuss this in the results section and refer to the supplementary data that shows these fold changes.
Overall comment about discussion section: I found this section highly speculative and think that the authors are trying to draw strong conclusions from one proteomics data set.
Lines 432-434: This is overreaching – the proteomic data only says that the protein level is increased or decreased, there is no experimental evidence to say that solute transport is actually increased
Lines 477-483: This is background information that would be more helpful to include when the bile salts experiments are introduced earlier in the paper
Lines 489-506: This seems too speculative – is there any evidence that this b-lactamase is important in antibiotic resistance? Metallohydrolases can have a variety of substrates. The NCBI annotation does not seem to suggest that this family of proteins has b-lactamase activity (https://www.ncbi.nlm.nih.gov/protein/WP_002288100.1)
Lines 550-552: This is true for some bacterial aggregation, but there are plenty of inter-species aggregation interactions that happen (many examples of oral bacteria) so it is incorrect to say that the presence of the same species is what triggers aggregation
Lines 570-573: this is highly speculative as many QS signals have to be taken up by specific transporters, not necessarily OppF. OppF is only one of the subunits of the Opp transport system.
Author Response
We thank the Reviewer 2 for the careful and precious evaluation of our work.
The figures' x axis has been modified as requested when appropriate.
Line 86: we possess stocks at -80°C, still, it is advisable to prepare agar plates or stocks at -20°C to start everyday experiments, and restart from the precious uncontaminated stock only once a month for example. At -20°C bacteria are perfectly conserved for short times.
Line 88: 5HT was freshly dissolved in CDM and diluted in CDM + 0,1 OD bacteria for the experiment. It has been added in current lines 96-97
Line 104-111: as stated in lines 109-111, the 5HT treatment occurred before harvesting. However, a little sentence has been included in line 112.
Line 125: we started with standard protocol (OD 0,1), but as our strain is not a great producer of biofilm, the amount of biomass stained was not adequate, and biofilm growth took long time, by which the molecule would have been degraded. Therefore, we decided to optimize the protocol seeding higher starting biomasses. Your considerations about gene expression are of true, still we always aim at a comparison between treated and untreated bacteria rather than a rigorous characterization of biofilm formation in enterococcus.
Figure 1: we thank the reviewer for these important considerations. We have performed the statistical analyses for final OD (24 hours) and also the CFU counting which will be reported as supplementary materials.
Figure 3: the figure has been modified into a bar char.
3.2.3 (Biofilm formation): actually, even when we were seeding bacteria at OD of 0.1 the biofilms obtained were not abundant. Our strain is not a great producer of biofilm, therefore, we hypothesize that the few cells that form biofilm between 0 and 24 hours are not sufficient to observe a stimulation induced by the molecule. However, after medium & 5HT replacement, and thus after removing almost all planktonic bacteria, we can affirm that the effect of molecules is direct on biofilms, and yes, of course this could be related to growth enhancement; still this could not be quantified by OD, as very few cells were left in suspension.
Lines 287-291: the method section is correct, therefore, the sentence in line 291 has been rewritten.
Figure 5: we agree with the reviewer that in a table ( new table 1) these results are more clear. We thank the reviewer for this valuable suggestion. Also, we represented Amp data before Van, as suggested.
Figure 6 (now Figure 5): we have modified the colors palette as suggested; still, we are persuaded that adding the gene loci at this point would be confusing. Everything is reported in the results and supplementary materials.
Figure 7 (now Figure 6): we apologize for the inconvenience and corrected it.
Overall comment about discussion section: the results’ section now contains gene loci and fold changes as requested by the reviewer.
Lines 432-434: Although enhanced transport activity has not been proved, in general, biological systems are regulated to avoid useless biochemical synthesis if the protein is not functionally required. Thus, we hypothesize that higher amounts of a certain enzyme/protein corresponds to increased function of this enzyme/protein. However, the sentence has been slightly modified.
Lines 477-483: in our opinion the considerations on the role of bile stress are more suitable here than in the methods since they better explain the following paragraph.
Lines 489-506: we agree with the reviewer for this consideration. However, as reported in the material section, the role of the protein has been hypothesized with several analyses, among which the use of uniprot database. In this database the function of the protein and the family can be addressed, with reference to other databases. InterPro, Pfam, SMART confirm the B lactamase domain. Of course, we cannot be sure of the activity, still, with the addition of the antibiotic resistance test, we can hypothesize that the protein may be involved. As a matter of fact, we use the words ‘the proteomic seems to suggest’, which are quite mild, and not a strong statement.
Lines 550-552: actually, we are not saying that ‘the presence of the same species is what triggers aggregation’. The sentence you are mentioning (now line 558) is not referred to general ‘aggregation’, which can occur both between cells of the same species and between not related organisms. We specifically mentioned ‘autoaggregation’, defining that it refers only to bacteria belonging to the same species.
Lines 570-573: we agree with the reviewer on this aspect. It should be reminded that we analyzed only the soluble proteome retrieving OppF, which could be only weakly associated with membranes and therefore detected. also, we are not stating that OppF is related to QS, we report just that this might be correlated to it. However, we rephrased the sentence to modulate the claim of our discussion.
Reviewer 3 Report
Dear authors,
My sole concern for your manuscript is the decision why and how the LAB E. faecium was chosen for this study. Especially in the introduction part (lines 75-77) should include an adequate justification.
Also consider the following:
Line 58: canalized - better use the work "channeled"?
Line 664: the acronym "NE" I believe appears for the first time and should be reported elsewhere where appropriate. In any case the abbreviation is not explained and is confusing. Apologies if I missed some information.
Author Response
We kindly thank Reviewer 3 for the suggestions, and we modified the text accordingly.
1) in line 58 canalized has been replaced by channeled, as required.
2)for line 75-77 a brief sentence has been added to explain the choice of the enterococcal strain
3)the acronym "NE" for "norepinephrine has been shown in the introduction section, when mentioned for the first time, and modified in the rest of the paper.
Round 2
Reviewer 2 Report
Figure 1 has OD595 as the y-axis label – is this data OD595 or OD600, as described in the text? Could the SM3 figure legend be modified to clarify that these are fold change values? Reading SM3 by itself, one could assume tha they are OD600 values.
Line 263-264 says CFU data is from 4 hr but the supplementary figure legend says the data is from 24 hr.
Supplementary data: how were these analyzed? T-test or ANOVA? Only T-test is described in Methods, but ANOVA needs to be used for >2 samples (such as the supplementary figures).
The authors did not directly address my concern about the data presented in Figure 3 – did the original version calculate statistical significance from technical replicates? Or from the biological replicates (which would be the more appropriate option)? Converting Figure 3 to a bar graph hides this even further.
Figure 4: units for antibiotic concentrations need to be added to this table
Author Response
We thank the reviewer for this second valuable exam of our article.
Figure 1: We thank the reviewer for detecting this imprecision. Now the result section (lines 258 and 261) is correct. The SM3 has been modified as requested.
Line 263-264: The text has been corrected.
Regarding SM2, each protein abundance in one condition or the other has been assessed with a T-test. With the same rationale, in the two other data sets of SM3, we compare each stimulation to the control condition with a T-test. Indeed, we are interested in determining statistical significance towards the untreated condition, while the comparison between the different concentrations of the molecule goes beyond the scope of this work. Also, the concentration of 50 µM 5HT has been retrieved for further investigations among the best effective over control conditions since it mirrors a possible pathological condition in the gut. However, if the reviewer considers these analyses as fundamental, we will eagerly provide them.
Figure 3: There has been an error in the previous data, and technical replicates data from two experiments have been presented. The picture now has been replaced with the correct one: the media of the 10 technical replicates has been considered as one biological replicate. The graph shows the media of 3 biological replicates, and of course, the statistical calculation has been performed on these 3 sets of data.
Figure 4: The units have been added to the table.
Round 3
Reviewer 2 Report
I thank the authors for their revision.
Regarding statistical tests, one-way ANOVA with comparisons to a control group (untreated) would be the most appropriate statistical test for the supplementary data as opposed to multiple t-tests. Instructions for this can be found in the Graphpad user manual. However, I do not believe that re-analyzing the data using ANOVA vs t-test would significantly change the biological conclusions of the paper.
Author Response
-